# Intercomparison of Same-Day Remote Sensing Data for Measuring Winter Cover Crop Biophysical Traits

**DOI:** 10.3390/s24072339

**Published:** 2024-04-06

**Authors:** Alison Thieme, Kusuma Prabhakara, Jyoti Jennewein, Brian T. Lamb, Greg W. McCarty, Wells Dean Hively

**Affiliations:** 1Sustainable Agricultural Systems Laboratory, U.S. Department of Agriculture-Agricultural Research Service, Bldg 001, BARC-W, 10300 Baltimore Avenue, Beltsville, MD 20705, USA; jyoti.jennewein2@usda.gov; 2Department of Geographical Sciences, University of Maryland, 2181 Samuel J. LeFrak Hall, College Park, MD 20742, USA; kusumaprabhak@gmail.com; 3U.S. Geological Survey, Lower Mississippi-Gulf Water Science Center, 2045 Route 112, Bldg 4, Coram, NY 11727, USA; blamb@usgs.gov; 4Hydrology and Remote Sensing Laboratory, U.S. Department of Agriculture-Agricultural Research Service, Bldg 007, BARC-W, 10300 Baltimore Avenue, Beltsville, MD 20705, USA; greg.mccarty@usda.gov; 5U.S. Geological Survey, Lower Mississippi-Gulf Water Science Center, Bldg 001, BARC-W, 10300 Baltimore Avenue, Beltsville, MD 20705, USA; whively@usgs.gov

**Keywords:** proximal sensors, satellite imagery, cover crops, biophysical traits, surface reflectance, biomass, fractional green cover, NDVI

## Abstract

Winter cover crops are planted during the fall to reduce nitrogen losses and soil erosion and improve soil health. Accurate estimations of winter cover crop performance and biophysical traits including biomass and fractional vegetative groundcover support accurate assessment of environmental benefits. We examined the comparability of measurements between ground-based and spaceborne sensors as well as between processing levels (e.g., surface vs. top-of-atmosphere reflectance) in estimating cover crop biophysical traits. This research examined the relationships between SPOT 5, Landsat 7, and WorldView-2 same-day paired satellite imagery and handheld multispectral proximal sensors on two days during the 2012–2013 winter cover crop season. We compared two processing levels from three satellites with spatially aggregated proximal data for red and green spectral bands as well as the normalized difference vegetation index (NDVI). We then compared NDVI estimated fractional green cover to in-situ photographs, and we derived cover crop biomass estimates from NDVI using existing calibration equations. We used slope and intercept contrasts to test whether estimates of biomass and fractional green cover differed statistically between sensors and processing levels. Compared to top-of-atmosphere imagery, surface reflectance imagery were more closely correlated with proximal sensors, with intercepts closer to zero, regression slopes nearer to the 1:1 line, and less variance between measured values. Additionally, surface reflectance NDVI derived from satellites showed strong agreement with passive handheld multispectral proximal sensor-sensor estimated fractional green cover and biomass (adj. *R*^2^ = 0.96 and 0.95; RMSE = 4.76% and 259 kg ha^−1^, respectively). Although active handheld multispectral proximal sensor-sensor derived fractional green cover and biomass estimates showed high accuracies (*R*^2^ = 0.96 and 0.96, respectively), they also demonstrated large intercept offsets (−25.5 and 4.51, respectively). Our results suggest that many passive multispectral remote sensing platforms may be used interchangeably to assess cover crop biophysical traits whereas SPOT 5 required an adjustment in NDVI intercept. Active sensors may require separate calibrations or intercept correction prior to combination with passive sensor data. Although surface reflectance products were highly correlated with proximal sensors, the standardized cloud mask failed to completely capture cloud shadows in Landsat 7, which dampened the signal of NIR and red bands in shadowed pixels.

## 1. Introduction

Winter cover crops have proven to be an effective method to reduce sedimentation and nutrient runoff into waterways, and increase environmental benefits including improved soil health and water quality [1,2]. As they accumulate biomass, winter cover crops sequester residual soil nutrients, and also prevent wind and water erosion of soils by providing vegetative groundcover [3,4]. They also have climate mitigation potential through carbon sequestration [5,6] and greenhouse gas reduction for non-leguminous species [7]. In ecologically sensitive estuaries such as the Chesapeake Bay, cover crop-mediated reductions in nitrate losses and sediment from agricultural areas have led to reduced risk of algal blooms, eutrophication, and threats to wildlife [2,8,9]. 

The U.S. Environmental Protection Agency (EPA) has identified winter cover crops as an effective practice to reduce agricultural pollutants and meet total maximum daily load targets (the amount of pollutant that a water body can safely assimilate) in the Chesapeake Bay [10]. Cover crop adoption in the Chesapeake Bay watershed is incentivized through national programs such as the U.S. Department of Agriculture (USDA) Natural Resources Conservation Service (NRCS) Environmental Quality Incentives Program [11] and through state-led initiatives such as the Maryland Department of Agriculture’s Maryland Water Quality Cost-Share Program [12]. Although cost-share payment amounts and enrollment vary each year, Maryland had a record of 561,344 acres of cover crops covering 64% of cropland planted in 2016 [1] and leads the nation in overall cover crop adoption [2,13]. 

Multispectral remote sensing has been demonstrably successful as a tool to monitor winter cover crop growth and biophysical traits such as biomass and groundcover [14,15,16,17,18,19,20,21] from both proximal (in-field or handheld) [17,21,22] and spaceborne platforms [16,18,19,20,21]. Satellite data are often publicly accessible, but there are tradeoffs between their temporal and spatial resolution. For example, in Talbot County, Maryland, fields average 20 hectares compared with 65 hectares in the Midwest [23]. This smaller size disallows use of coarse-resolution satellites with high temporal resolution such as the Moderate Resolution Imaging Spectroradiometer (MODIS; 250 m) that acquire imagery daily. Satellite imagery with fine to moderate spatial resolution (2 m to 30 m pixel size) are more appropriate for measuring small fields typical in the Chesapeake Bay watershed but are still subject to issues related to mixed pixels, particularly in fall and winter when cover crop canopies are not fully closed and soil and crop residues are present. Many of these platforms such as Landsat 7 (30 m), Sentinel-2 (10–30 m), Satellite pour l’Observation de la Terre 5 (SPOT 5; 10 m), and WorldView-2 (2–5 m) are passive sensors that can only collect accurate data over areas free of clouds and cloud shadows [24,25]. Because of these limitations, passive satellite platforms may experience large data gaps due to return frequency, cloud cover, and snow presence, particularly during the winter cover crop season (October through May). These spatiotemporal shortcomings in fine-to-moderate scale imagery may be overcome if integrated with each other and proximal instrumentation. 

Proximal reflectance sensors operate at a short distance from the ground (0.1–5 m) and can be used to quantify biophysical traits and their in-field variation at very fine spatial scales (~2 cm) [26,27,28,29]. These sensors can be handheld, pole mounted, or mounted on tractors [17,21,26,27,28,29,30]. Proximal sensors record reflectance measurements with minimal interference from the atmosphere, theoretically providing accuracy assessment comparisons with atmospherically corrected satellite imagery. 

Both satellite and proximal sensors can be passive, collecting reflected sunlight, or active, emitting light and collecting the reflectance of the emitted light. Passive sensors are designed to be collected near solar noon at a similar time of day for each successive collection. Since active sensors emit their own light, they can be operated in varied light conditions and handheld proximal active sensors can be operated under cloudy conditions. Active sensors also use more operational energy and are more common in proximal or airborne platforms than satellite platforms. Although several studies compare the interchangeability of passive satellite sensors [31,32,33], the ability to estimate biophysical traits using proximal active and passive sensors [34,35,36,37,38], satellite passive and uncrewed aerial vehicle passive sensors [39,40], or a passive satellite sensor with passive proximal sensor [41], there is currently a gap in research comparing multiple passive satellite sensors with proximal active and passive sensors. Additionally, most sensor intercomparison studies focus on summer crops with higher biomass conditions rather than detecting green biomass under the low biomass conditions associated with monitoring cover crop performance. Exploring the relationship between high-spatial resolution handheld sensor data and satellite-based measurements can allow for scaling up of difficult-to-acquire proximal data, resulting in greater operational efficiency when assessing the environmental benefits of cover crop implementation. 

In this study we sought to: (1) quantify variation between and among three passive satellites (Landsat 7, WorldView-2, and SPOT 5) with two processing levels (surface reflectance [SR] and top-of-atmosphere reflectance [TOAR]) and coincident handheld proximal sensors (one passive and one active) red and near infrared (NIR) band reflectance and normalized difference vegetation index (NDVI) data, and (2) quantify variation in sensor-derived estimates of cover crop biophysical traits (biomass and fractional groundcover) attributed to the various sensors. Although previous studies have explored the relationship between satellite reflectance measurements and SR measurements from stationary ground-based instruments, this research compares ground-based “on-the-go” proximal sensor data with fine and moderate resolution satellite imagery that have been corrected to both TOAR and SR. The TOAR and SR processing levels, in addition to SR processing tools, have been shown to exhibit differences in band reflectance and NDVI values [42,43]. Thus, TOAR vs. SR comparisons are also a key focus of this research effort. This research has the potential to evaluate the interchangeability of commonly used satellite and proximal sensor NDVI with widely available processing levels for estimating cover crop biophysical traits. 

## 2. Materials and Methods

### 2.1. Study Site and Design

All data were collected on fields located at the U.S. Department of Agriculture-Agricultural Research Service (USDA-ARS)—Beltsville Agricultural Research Center (BARC) in Beltsville, Maryland, USA during the winter of 2012–2013. Cover cropped fields sampled in this study (*n* = 5) included two barley (*Hordeum vulgare* L.) fields, one ryegrass (*Lolium multiflorum* Lam.) field, one triticale (*Triticale hexaploide* Lart.) field, and one wheat (*Triticum estivum* L.) field, all of which are commonly planted cover crop species in the Mid-Atlantic region (Figure 1). Planting dates and management methods are described in Prabhakara et al. [17].

Within each field we collected satellite, handheld sensor, and biophysical data including biomass, fractional groundcover, carbon concentration, and nitrogen concentration. We used a combination of data from three passive multispectral satellites (Landsat 7, SPOT 5, and WorldView-2) with different pixel sizes and resampled the pixel size to the largest of the three, 30 m × 30 m. Field sampling occurred on two dates: 6 December 2012, and 23 January 2013 (Figure 2). On the same dates and at overlapping times near midday we collected proximal sensor data from an active sensor, Crop Circle, and a passive sensor, CROPSCAN in tracks that passed through the interior of each sampled Landsat pixel [44,45]. Additionally, one in situ biomass sample (0.5 m^2^) and three nadir RGB photos were collected near the centroid of each Landsat pixel (Figure 2). 

### 2.2. Sensors

In total, five different sensors were used in this analysis, which included three satellites (Landsat 7, SPOT 5 and WorldView-2) and two proximal sensors (passive CROPSCAN and active Crop Circle) [44,45]. The three satellite platforms have different temporal, radiometric and spatial resolutions as described in Table 1. The handheld proximal sensors were GPS-enabled to collect measurements every two to three seconds as they moved across the landscape. Satellite imagery pairs were acquired on 6 December 2012 (Landsat 7, WorldView-3) and 23 January 2013 (Landsat 7 and SPOT 5), with proximal sensor data collected simultaneously (Table 1).

The atmospheric conditions on both days were evaluated using AERONET data [46] from the Goddard Space Flight Center in Greenbelt, Maryland. Atmospheric visibility was calculated with a log-linear interpolation of aerosol optical thickness (AOT) at 500 nm and 667 nm to derive AOT at 550 nm which was used to estimate visibility. Both days were very clear, but 6 December 2012 (300 km visibility) more so than 23 January 2013 (157 km visibility) [46]. On 23 January, there were popcorn clouds scattered throughout the image (Figure 3), whereas no clouds or shadows were present on 6 December. The high visibility levels indicate clear conditions, providing adequate irradiance to the proximal sensors.

#### 2.2.1. Landsat 7

Atmospherically corrected Landsat 7 SR and TOAR scenes from 6 December 2012 and 23 January 2013 (path 15/row 33) were downloaded from the U.S. Geological Survey (USGS) Earth Resources Observation and Science (EROS) Center Science Processing Architecture [47] in 2013. Landsat 7 imagery were processed to SR with the Landsat Ecosystem Disturbance Adaptive Processing System (LEDAPS) algorithm [48]. This processing occurred prior to downloading from Earth Explorer [49]. Although Landsat 7 had scan line errors over a significant portion at the edges of its swath, our sampling locations were unaffected by this issue because they fell in the center of the swath [50]. 

Cloud masking of Landsat was achieved by applying the F mask band of Collection 2 [24]. The F mask identifies potentially cloud-, water-, snow-, or ice-covered pixels based on their spectral and thermal properties and pairs clouds with cloud shadows using object-based matching; these pixels are identified in the F mask band and can be removed from analyses [45]. The 23 January Landsat 7 image contained cloud shadows covering 17 sampling points within the triticale and the first barley fields that were not detected by the cloud mask (Figure 3). Cloud shadows over the study sites were only present in the Landsat 7 image due to differences in imaging times (the Landsat 7 image was acquired at 15:52:38 GMT and the SPOT 5 image at 16:05:22 GMT). Cloud shadows are typically removed as “contaminated” pixels that are not useful for derivation of surface characteristics [51]. Because the cloud shadows affected reflectance, statistical models using Landsat 7 imagery were calculated with the shadowed sampling points removed (*n* = 17). Cloud shadowed Landsat 7 pixels were removed from the Landsat 7 comparisons but were maintained in all other sensors. 

#### 2.2.2. SPOT 5 and WorldView-2

A SPOT 5 image for BARC (path 622/row 271) was tasked as part of the USGS North American Data Buy, acquired on 23 January 2013, and downloaded from USGS Earth Explorer [49]. SPOT 5 features a 10-m spatial resolution and four bands (Figure 3). WorldView-2 imagery was acquired on 6 December 2012 (Imagery copyright 2012 Maxar, https://maxar.com/maxar-intelligence/products/satellite-imagery [accessed on 8 July 2023], Westminster, CO, USA). The WorldView-2 sensor has 8 spectral bands (Figure 3) with a 2-m spatial resolution for visible and NIR bands. The raw SPOT 5 and WorldView-2 images were manually georegistered using a linear pixel shift to match a field boundary polygon vector shapefile and ensure proper alignment by visual inspection in ENVI version 4.8 [52]. TOAR was calculated in ENVI using image metadata. SPOT 5 and WorldView-2 imagery were converted to SR using MODTRAN 5.3.3. radiative transfer code [53,54]. MODTRAN has generally been found to closely intercompare with other atmospheric correction tools including FLAASH and Sen2Cor [43,55] as well as 6S [56,57]. 

MODTRAN parametrization (input) data required for atmospheric correction include information about various aerosols (i.e., AOT at 550 nm [AOT550]), water vapor (total column values in cm), and ozone (total column values in Dobson units). AOT550 and water vapor values were derived from AERONET data from the Goddard Space Flight Center site [46]. Total ozone data were estimated using Environment Canada daily ozone maps [58]. MODTRAN vertical profiles were established using radiosonde data that included temperature and relative humidity measurements in the upper atmosphere, which were acquired for the Dulles Airport/IAD site as it was the closest station in a peri-urban area [59]. To allow for a more precise comparison against other sensors, the SPOT 5 and WorldView-2 imagery were smoothed using a low pass filter and a 3 × 3 (SPOT 5) and 15 × 15 (WorldView-2) kernel to more closely match Landsat 7’s spatial resolution. 

#### 2.2.3. CROPSCAN

The CROPSCAN MSR16R handheld passive multispectral radiometer [44] used in this study was able to record “on-the-go” measurements every 3 s as it crossed the landscape by deriving geographic coordinates associated with each reflectance data point from a Trimble GeoXH (Westminster, CO, USA) GPS unit with sub-meter accuracy. CROPSCAN has a greater spectral resolution than either Landsat 7 or SPOT 5, gathering data across 16 spectral bands that are centered on the visible-near infrared (VNIR) portion of the electromagnetic spectrum. The CROPSCAN instrument was mounted on a hand-held pole with nadir view angle, approximately 1.8 m above the vegetation canopy, creating a 1 m^2^ field of view. The instrument employs a two-way sensor system that measures incident (downwelling) solar irradiance and upwelling radiance from the ground, enabling self-calibration to SR. When illumination conditions are not optimal due to cloudy conditions or low sun angles, lack of radiation reaching the sensor can result in inaccurate SR measurements. Therefore, all data with downwelling irradiance values below 300 W m-22 were excluded from analysis. All collections occurred under clear sky conditions between 10 a.m. and 2 p.m. solar standard time to avoid signal:noise reductions associated with low sun angles [60]. CROPSCAN collections (*n* = 2740, average 39 scans per sample; Table 1) were averaged within each Landsat 7 pixel (buffered inward 5 m to reduce edge effects) to match spatial resolution between the sensors. 

#### 2.2.4. Crop Circle

A Crop Circle ACS-430 active proximal sensor [45] was deployed in tandem with the CROPSCAN to gather spectral data. The Crop Circle is an active sensor with its own illumination source, and therefore has fewer sky-condition limitations than passive radiometric sensors which require specific solar illumination conditions. However, active sensors can result in low-biased or high-biased reflectance readings when compared with passive satellite sensors [35,61,62]. Active remote sensing signals can also be affected by plant canopy characteristics, distance from the plant canopy, and device temperature [35,63]. The spectral resolution of the Crop Circle contains three bands in the red, red-edge, and NIR portions of the electromagnetic spectrum. The sensor was held approximately 1 m above the soil surface creating a 0.13 m^2^ field of view and collected approximately one scan per second, associated with a GPS coordinate. Crop Circle readings (*n* = 31,497; average 450 scans per sample; Table 1) were collected and averaged within each buffered (−5 m) Landsat 7 pixel. 

#### 2.2.5. RGB Photography and Destructive Biomass Samples

At 23 of the 35 sensor sample locations, we collected three shoulder-height (1.5 m) red-green-blue (RGB) photographs using a Nikon D3100 DSLR camera (Minato City, Tokyo, Japan) repeated on two dates (14 December 2012, and 23 January 2013) between 10:00 a.m. and 2:00 p.m. solar standard time (Figure 1 and Figure 2, Table 1). Photographs (*n* = 138; 23 samples × 3 photos per sample × 2 dates) were processed using SamplePoint v1.60 software [64] to quantify fractional green vegetation, chlorotic yellow vegetation, and crop residue cover. A smaller number of photos were collected alongside the sensor collection on 6 December 2012 (10 sampling locations, *n* = 30 photos). A comparison of photos 6 December and December 14 shows only a small difference (average difference +/− 4% green vegetation) between the dates. SamplePoint randomly placed 200 points within each photograph and the groundcover (S = soil, GV = green vegetation, R = residue, O = other) under each point was recorded and subsequently summarized to derive the percent groundcover for each photo. The following equation was used to estimate fractional green cover from SamplePoint data: Fractional Green Cover = GV/(S + GV + R + O) × 100(1)

We then took the mean of the three photographs per pixel. 

At the same 23 photo sampling locations, biomass samples (*n* = 46) were collected within 3 m of each Landsat pixel centroid by cutting a 1 m length of 3 adjacent rows of cover crop at ground height (0.5 m^2^ surface area). Samples were dried for 48 h at 60 °C and weighed. The sampling area and dry weights were extrapolated to estimate biomass at the field scale (kg ha^−1^). Although biomass samples were not measured on the same day as sensor data collection, they were collected within two weeks of the sensor dates and <25 growing degree days (GDD) accumulated between sampling and imagery acquisition. It was therefore assumed that biomass, percent vegetative groundcover, and composition of the samples would remain relatively static due to minimal cover crop growth during the cold weather conditions (Figure 4). 

Because we collected RGB photographs and destructive biomass samples at fewer locations than proximal sensor collections, we first compared CROPSCAN derived biomass and fractional green cover to the physical sample data (Appendix A). Then, because we observed very strong relationships for both biophysical traits, we used the passive handheld sensor-derived biophysical traits to compare as the “gold standard” to the remaining sensors [65]. 

To assess cover crop biophysical traits we used NDVI [66], which is defined as: NDVI = (NIR − Red)/(NIR + Red)(2)
in combination with existing calibration equations for fractional green cover and biomass [18], which have been shown to be highly correlated with winter cover crop biophysical traits prior to index saturation [17,18,19]. Index saturation occurs as red reflectance has little variance in moderate to high biomass plants while NIR reflectance increases in higher biomass plants creating small or no increase in NDVI with increased biomass beyond saturation [67,68]. The saturation point may vary across active and passive sensors as reflectance values can vary with different light sources [38]. For each of the five sensors, we calculated fractional green vegetation cover and biomass using the following equations:fractional green cover = −21.904 + 116.305 × NDVI(3)
ln(biomass) = 3.2022 + 5.3740 × NDVI(4)
which were developed for cereal cover crop samples with Landsat 5, Landsat 7, Landsat 8, and Harmonized Sentinel-2 SR NDVI imagery [18]. While indices improving on NDVI have been proposed, many of these indices are based on the same fundamental visible to NIR relationship as NDVI [67,69,70] and comparison of 9 multi-spectral indices showed NDVI to be a top performer [17,21].

### 2.3. Growing Degree Days

Growing degree days (GDD) predict the timing of phenological milestones in plants [65]. The base temperature is set to a point where plants are unable to grow (Tbase), which varies based on species. GDD were calculated using the formula:GDD = [Tmax + Tmin)/2] − Tbase(5)
where Tmax and Tmin are daily maximum and minimum temperatures, and Tbase is 4° Celsius, a common base value for the winter cereal species included in this study [65]. Daily minimum and maximum were obtained from the National Oceanic and Atmospheric Administration (NOAA) National Centers for Environmental Information Daily Summaries Station located at Beltsville, MD, USA [66]. Daily mean temperatures below Tbase were set to Tbase, and then base temperature was subtracted from this value. 

The temperature during the sampling period was slightly above the climate normals (10-year average) for the area (Figure 4). GDD accumulation from 15 November indicates the relative warmth of the cover crop growing season (Figure 4, light gray). There was no snow on the ground during either sampling date. In the 48 days between the sampling dates mild freezing occurred, and a total of 70 GDD were accumulated (Figure 4, dark gray). Importantly, <20 GDD accumulated and one freezing event occurred between the 6 December 2012, sensor collection and the 14 December 2012 biomass sampling. There were also <20 GDD accumulated between the 10 January 2013, biomass sampling and 23 January 2013, sensor collection.

### 2.4. Statistical Analysis for Sensor Intercomparison

We used simple linear regression models for sensor-to-sensor comparison of red, NIR, and NDVI values as well as for NDVI-calibrated fractional green vegetation cover and biomass comparisons (Equations (2) and (3)) using the R Software Program v2023.09.1+494 [71]. For each model we compared goodness of fit and error using the coefficient of determination (adj. *R*^2^) and the root mean square error (RMSE). To compare how estimates of cover crop biophysical traits varied between sensors, we used slope and intercept contrasts in the “emmeans” package [72]. 

To better compare NDVI values among sensors, we used passive handheld sensor measurements as the “gold standard” as it is a self-calibrating proximal, sensor with upward and downward facing sensors that account for incoming and reflected radiation and can be used in partially cloudy conditions [44,65]. We compared the absolute value of differences between reflectance values of each sensor and the CROPSCAN measurements using the following equation:differenced NDVI = |(Sensor NDVI − CROPSCAN NDVI)|(6)

This reduces the dimensionality of the dataset that is present due to large biomass and NDVI variability among fields. Finally, we used analysis of variance (ANOVA) to test for statistical differences among these absolute value NDVI estimates. We employed Tukey’s post-hoc test to determine which sensors had statistically different NDVI values using a 95% confidence (*p* < 0.05).

## 3. Results

This study analyzed the comparability of red and NIR band and NDVI values among several remote sensing measurements taken on the same day at similar times (Table 1). On both 6 December 2012 and 23 January 2023, mid-day Crop Circle, CROPSCAN, RGB photos, and Landsat 7 data were collected (Table 1). Additionally, 6 December 2012 featured WorldView-2 data and 23 January 2013 featured SPOT 5 data. 

### 3.1. Reflectance Comparisons of Satellite and Proximal Sensors

We compared satellite SR data to passive proximal sensor data CROPSCAN for two bands (red and near infrared) that had similar bandwidths across all sensors. Correlation between satellites and CROPSCAN red reflectance resulted in high accuracies (adj. *R*^2^ ≥ 0.89 for all satellites) for both December and January samplings (Table 2). Within these comparisons, we observed that satellite red bands overpredicted red reflectance of CROPSCAN (slopes ranged from 1.08 to 1.187). Similarly, red reflectance between satellites was highly correlated, with goodness of fit values between Landsat 7 and WorldView-2 and Landsat 7 and SPOT 5 of 0.96 for both dates (Table 2). 

For the near infrared bands, we observed high accuracies between satellites and CROPSCAN proximal reflectance (adj. *R*^2^ = 0.94 for Landsat 7; adj. *R*^2^ = 0.95 for Worldview) for SR imagery on 6 December (Table 2). However, these relationships were weaker between CROPSCAN and Landsat 7 on 23 January (adj. *R*^2^ = 0.79) as several points (*n* = 17) were eliminated in the Landsat 7 analysis due to cloud shadows, reducing the overall number of sampling points available for comparison, in addition to the possibility that cloud shadow effects may have been present in non-eliminated points. As with the red bands, satellite NIR reflectance values were highly correlated to each other, with goodness of fit values between Landsat 7 and WorldView-2 of 0.97 in December and goodness of fit values between Landsat 7 and SPOT 5 in January of 0.98 (Table 2). When sampling points under cloud shadows were not eliminated from analysis, relationships between Landsat 7 SR and CROPSCAN degraded considerably, with goodness of fit (*R*^2^) falling from 0.90 to 0.86 for red bands and 0.94 to 0.21 for NIR bands. 

Additionally, satellite TOAR and SR band and NDVI values were measured against one another and against proximal sensors (Table 2 and Table 3, Figure 5). Landsat 7 and WorldView-2 NDVI had very high goodness of fit with CROPSCAN, 0.95 and 0.99, respectively (Table 2, Figure 5). In both cases the slope of the line deviated only slightly, and the y-intercept was close to zero. The relationship between Landsat 7 and WorldView-2 NDVI values were highly correlated with an adjusted *R*^2^ of 0.97 and slope and intercept near 1:1 (Table 2). Crop Circle was also highly correlated with satellite reflectance values (*R*^2^ of 0.90 to 0.94) but showed consistently low-biased reflectance values as seen invariable slopes and the large-magnitude intercept values (Table 2, Figure 5). Passive measurements of NDVI were prone to saturation above 0.8 (Figure 5) while the active sensor (Crop Circle) NDVI values were more linear (Figure 5G). 

### 3.2. Relation of Satellite Processing Level and Proximal Sensors

We also assessed how SR and TOAR corrections varied among satellites and the coincident proximal sensor collections. Overall, we observed that satellite SR measurements were more tightly clustered with CROPSCAN than with Crop Circle. Crop Circle NDVI had the largest deviations in measurements, with values that were consistently below the 1:1 line (Figure 5 and Appendix A). We also observed higher SR NDVI compared to TOAR due to increased reflectance in the NIR and decreased reflectance measurements in the red band (Table 2 and Table 3, Figure 5). As with individual bands relationships in the visible and near-infrared, *R*^2^ values for NDVI were identical between satellite TOAR and SR, but slope and intercept values varied slightly with NDVI underpredicted in satellite TOAR (Table 3, Figure 5). 

ANOVA post-hoc tests were used to evaluate the calculated absolute value differences in NDVI between CROPSCAN values (i.e., the “gold standard”) and the other sensors (Figure 6) collected on both sampling dates. The Landsat 7 TOAR from cloudy points showed the greatest deviation from CROPSCAN and from other sensors (Figure 6). Similarly, Crop Circle showed one of the larger deviations from CROPSCAN NDVI and from the other sensors and processing levels. The difference in NDVI between CROPSCAN and Crop Circle reduced as Crop Circle NDVI values increased, with average differences above NDVI = 0.8 of only 0.06. This may indicate delayed NDVI saturation of Crop Circle compared to CROPSCAN. Landsat 7 SR Cloudy differenced NDVI also varied significantly from non-cloudy Landsat 7 SR points. We observed no statistical differences between SR NDVI values derived from any of the three satellites. Landsat 7 TOAR clear observations and WorldView-2 did not statistically vary in differenced NDVI, but SPOT 5 TOAR differed from both other satellite sensors. 

### 3.3. Sensor Intercomparisons of Cover Crop Biophysical Traits

December cover crop fractional green vegetative cover varied between 8 and 99 percent, and January collections varied between 18 and 98 percent. This wide range of fractional green vegetative cover values provided a suitable basis for evaluation of proximal- and satellite-derived estimates of fractional cover. We then used Equations (3) and (4) to derive satellite estimates of fractional green cover and biomass and compared the resulting values. Results from our comparisons of fractional green cover from SamplePoint and CROPSCAN were highly accurate with low error (adj. *R*^2^ = 0.96, RMSE = 5.52%; Appendix A, SF2). CROPSCAN also had the strongest relationship with in situ biomass of any sensor considered in this study (Appendix A SF3). In this section, we only use SR and not TOAR imagery as comparisons between SR and TOAR were made in the preceding section. 

Results from our fractional green cover models (Figure 7) demonstrated that all sensors produced high accuracies (adj. *R*^2^ = 0.95–0.99) and low errors (RMSE = 2.37–4.76%). All SR slopes slightly overpredicted fractional green cover (slopes = 1.09–1.42), while Crop Circle slope underpredicted fractional green cover (slope = 0.87). We observed that SPOT 5 SR had statistically lower intercepts than either Landsat 7 SR (T-ratio = −2.95, *p* = 0.02) and WorldView-2 SR (T-ratio = −4.29, *p* = 0.0002). Importantly, we observed that Crop Circle intercepts were consistently much larger than any other sensor resulting in underprediction of fractional green cover when using Equation (2) (T-ratios = 11.90–17.64, *p* < 0.0001) and implying that Crop Circle would require development of a separate calibration to successfully measure fractional green cover. There was also a deviation in the Crop Circle estimations above 70% also noted in the NDVI comparison above NDVI of 0.75 (Figure 5G and Figure 7). This may indicate that the passive sensors (CROPSCAN, Landsat, SPOT, WorldView) saturate at a lower NDVI value than the active sensor (Crop Circle).

We also collected destructive biomass samples for a subset of sampled points (*n* = 36), which ranged from 75–4202 kg ha^−1^. Results from our comparisons of destructively sampled biomass and CROPSCAN estimated biomass had a high accuracy (adj. *R*^2^ = 0.80) and low error (RMSE = 0.05 or the equivalent of ~56 kg ha^−1^; Appendix A), when applying logarithmic scaling. Relationships when using raw biomass values on a non-logarithmic scale, demonstrated previously identified saturation issues ~1500 kg ha^−1^ [17,21] and as a consequence had lower accuracy and higher error when compared to CROPSCAN derived biomass (adj. *R*^2^ = 0.60; RMSE = 1123 kg ha^−1^). Therefore, to increase the number of data points for comparison to other sensors we employed CROPSCAN derived biomass as our outcome (or “truth”) variable in the comparisons below. 

Results from our de-logged biomass models (Figure 8) demonstrated that all sensors and processing levels had high accuracies (adj. *R*^2^ = 0.96–0.99) and low errors (RMSE = 114.36–351.43 kg ha^−1^). SPOT 5 and WorldView-2 slopes (3.31 and 2.87, respectively) slightly under-predicted biomass and were not statistically different from one another. Similar to fractional green cover results, we observed that Crop Circle intercepts were consistently larger than any other sensor resulting in under-prediction of biomass (T-ratios = 6.88–9.21, *p* < 0.0001). Again, we note a deviation from the linear relationship at high biomass with Crop Circle, a difference expected as NDVI saturation occurs in CROPSCAN earlier than Crop Circle (Figure 8). SPOT 5 intercepts were consistently higher than other sensors resulting in potential over-prediction of biomass and fractional groundcover when observed biomass or fractional groundcover is low (~<400 kg ha^−1^ or ~45%, respectively).

## 4. Discussion

Strong similarities were found between passive sensors (CROPSCAN, Landsat 7 SR, WorldView-2 SR, and SPOT 5 SR) in their spectral characteristics (Table 2; Figure 5), and their estimates of cover crop biophysical traits (Figure 7 and Figure 8). These results demonstrate that under clear sky conditions there is high correlation not only between ground-based proximal sensors and moderate- to fine-spatial resolution satellite platforms, but also high correlation among the satellite SR products. In practical terms, this indicates that a variety of passive multispectral satellites and proximal instruments provide comparable values, particularly for NDVI, which has important implications for quantifying and mapping winter cover crop biophysical traits. For instance, categorizing cover crops by the amount of biomass or fractional green cover is useful on a landscape scale, where high biomass accumulation provides greater water quality benefits compared to areas with lower biomass accumulation [15]. The calibration equations (Equations (3) and (4)) used to estimate cover crop biophysical traits from multispectral data were originally derived from a combination of in situ biomass samples, fractional green cover samples, and passive multispectral SR imagery from several moderate resolution satellites (Landsat 5, Landsat 7, Landsat 8, Harmonized Landsat Sentinel-2) [18]. Since WorldView-2 and SPOT 5 sensors have comparable red and near infrared bandwidths with the moderate resolution satellites used to develop the calibration equations (Table 1), we tested our ability to extend the application of the calibration equations to fine scale imagery from WorldView-2 and SPOT 5. Our results suggest that SR products from Landsat 7 and WorldView-2 could largely be used interchangeably to estimate cover crop biophysical traits, which could be useful in addressing spatiotemporal issues present in fine- to moderate-scale satellite platforms. Additionally, the newer WorldView-3 satellite features nearly identical red and NIR bands as WorldView-2 providing further WorldView-Landsat interoperability. However, although SPOT-5 imagery had similar accuracies, errors, and slopes to other spaceborne sensors used in this study, its intercept was significantly lower. This indicates that as observed groundcover and biomass are lower SPOT-5 estimates would be biased high (Figure 7 and Figure 8). 

In comparing individual bands, we found increased NIR reflectance measurements in SR compared to TOAR (average reflectance difference 0.018 SPOT, 0.015 Landsat, 0.259 WorldView-2). Red reflectance measurements were smaller for SPOT SR (average −0.015) and Landsat SR (−0.012) and slightly larger in WV SR (0.040) compared to TOAR. The combination of shifts in these bands led to higher NDVI measurements calculated from SR (average difference 0.041 SPOT, 0.034 Landsat, 0.025 WV) compared to TOAR, a shift typical in atmospheric corrections over vegetated areas [73] (Figure 5). These NDVI differences demonstrate that even for images with minimal haze/high visibility, atmospheric correction is critical for achieving accurate surface reflectance retrievals. The largest differences in band reflectance values was seen in Crop Circle, potentially as the active sensor can have band value differences that occur when the distance of the sensor changes relative to the target (cover crop) and was designed for NDVI measurements rather than single band values [35]. A correction would need to be applied to compare band reflectance values directly between Crop Circle and the passive sensors included in this study [35].

Although we observed very strong relationships between SR NDVI and derived biophysical traits and CROPSCAN, we noted several important differences between the active sensor (Crop Circle) and all of the passive sensors. Crop Circle NDVI values were consistently lower than any passive sensor regardless of processing level (Figure 5) which is similar to results found in previous comparisons of passive and active proximal sensor NDVI [35,36,37]. This may be due to a longer wavelength red band center in Crop Circle (670 nm) than the passive sensors (645–660 nm) which is the start of the red-edge portion of the electromagnetic spectrum (670–737 nm), an area noted for being more sensitive to plant quantity and health than the visible red (650–670 nm) [74,75]. The lower NDVI values in Crop Circle led to consistent under-predictions of cover crop fractional green cover (Figure 7) and biomass (Figure 8) using Equations (3) and (4), respectively. These results indicate that the existing equations for deriving cover crop biophysical traits from NDVI are not appropriate for Crop Circle without an intercept correction. Existing equations for deriving cover crop biophysical traits from NDVI draw from passive sensor data with slightly different band centers and are not appropriate for Crop Circle without an intercept correction [16,17,18,76]. Additionally, active remote sensing signals can be affected by collection rate, plant canopy characteristics, distance from the plant canopy, and device temperature [35,63]. The low NDVI values returned from Crop Circle are likely due to lower NIR signal returns compared with sunlight for the passive sensors. as seen in the large NIR slope difference between CROPSCAN and Crop Circle in Table 2. Crop Circle may be appropriate for isolated analysis as it is shown to have consistent relationships with fractional cover and biomass [35], but in this study did not have comparable SR measurements to passive proximal and space-based sensors. Therefore, these results suggest that new calibration equations would be needed for deriving fractional green cover and biomass from active spectral sensors such as Crop Circle, or, alternatively, estimating a robust intercept offset term. 

We also observed that the automated cloud and cloud shadow masks delivered with Collection-2 Landsat 7 SR products did not detect many of the popcorn clouds and shadows they cast in the 16 December image (Figure 4), indicating that visually identifying cloud shadows prior to analysis could be used as an additional quality check. While all cloud masking routines are prone to some error [25], improvements to cloud masks are expected for updated Collection-2 processing of Landsat 8/9 imagery [77] in addition to improved geometric and atmospheric correction [78] compared to Collection-2 Landsat 7 imagery used in this study. Sentinel-2, a moderate resolution spaceborne satellite, is prone to similar cloud mask errors as seen here, although errors are reduced when using the Harmonized Landsat Sentinel-2 (HLS) data product with improved cloud masking and atmospheric correction [79]. 

The observed red and NIR band differences in sampling points under cloud shadows can be explained due to differences in atmospheric scattering at various wavelengths. Clouds block and absorb nearly all near-infrared light and there is minimal scattering from the atmosphere to the surface leading to low reflectance values. Accordingly, the coefficient of determination (*R*^2^) between Landsat 7 and SPOT 5 fell from 0.98 to 0.15 for NIR bands from the shadowed pixels. In the shorter green and red wavelengths, increased atmospheric scattering, both from the haze in the atmosphere and diffuse radiation from the surface, can result in a fairly strong signal as scattered light fills in shadowed areas [24], explaining the lesser degree of degradation of the red shadowed reflectance (0.96 to 0.85) relative to the NIR shadowed reflectance. This is in accordance with research by Simpson and Sitt [80] demonstrating that cloud shadows exert a greater influence on near-infrared bands than visible bands. Although the effect is less pronounced in the relationship between Landsat 7 SR and CROPSCAN NDVI with goodness of fit (*R*^2^) falling from 0.95 to 0.92 and average absolute difference in NDVI increasing by 0.03, the effect is amplified when estimating biomass (RMSE increased from 259 kg ha^−1^ to 426 kg ha^−1^) or fractional cover (RMSE increased from 4.8 to 6.4%) derived from NDVI. If we only examine the sampling points under cloud shadows the RMSE of estimated biomass is 758 kg ha^−1^ and the RMSE of fractional cover is 9.8%. This emphasizes the potential impact of cloud mask errors when cloud shadows are not detected.

When satellite imagery was converted to SR and compared with proximal data, intercepts were closer to zero and slopes were closer to the 1:1 line than when TOAR imagery was used (Table 2 and Table 3). The exceptions were sampling points covered by cloud shadow as in the 23 January Landsat 7 image. When these points were included in the analysis, near infrared measurements showed a poor relationship between satellite and proximal sensors and between both satellite images. When the cloud shadow points were removed, correlations increased between Landsat 7 and passive proximal sensor readings from *R*^2^ of 0.91 to 0.95. These findings highlight the need for both atmospheric corrections along with high quality cloud and cloud shadow detection. NDVI between satellites and proximal sensors also exhibited high goodness of fit among sensors on both dates (Figure 5 and Figure 6). For the 23 January Landsat 7 imagery, conversion from TOAR to SR followed by calculation of NDVI resulted in a better match between satellite and proximal sensors. Although conversion to SR reduced some of the effects of cloud shadows on Landsat 7 NDVI, the effects were amplified in estimates of biophysical characteristics with doubled and tripled error estimates (RMSE) for fractional cover biomass, respectively, in the cloud shadow pixels compared to estimates from CROPSCAN. The impacts of cloud shadow were clearly demonstrated in this study. Further study of these impacts could focus on updates to cloud masking procedures comparing Collection-1 vs. Collection-2 processing for individual Landsat 7 sensors, and comparison of cloud masking procedures for Landsat 7 compared to Landsat 8 and 9 with different routines for cloud masking and atmospheric correction [48,78]. The inclusion of a cirrus band on Landsat 8–9 instruments has been demonstrated as beneficial for improving the accuracy of cloud screening [78,81] as has subpixel shift detection for sensors that lack thermal bands such as Sentinel-2 [82]. 

Atmospheric correction of TOAR satellite data to SR resulted in reflectance values that were closer to proximal SR than were TOAR data. This is important for categorizing cover crops and estimating biophysical traits based on their NDVI values. There was a high goodness of fit between percent vegetative groundcover and NDVI values for all proximal sensor and satellite measurements. Although the conversion to SR did not improve correlation to fractional cover for individual dates, the higher consistency for NDVI for SR processing is needed for time series applications. Conversion to SR is important when comparing across satellite sensors, particularly for commercial image sources like SPOT 5 and WorldView-2 that are often delivered in TOAR format.

Data in this study were collected under relatively clear sky conditions to ensure that adequate irradiance reached the sensors. To acquire clear satellite imagery and be able to capture ground conditions with proximal sensors, mostly clear conditions are necessary. SR data comparisons from satellites might be more important under conditions of low atmospheric visibility where there is increased atmospheric interference and calibration to ground-based sensors is more challenging. NDVI in particular is impacted by aerosol effects if they are not properly accounted for through atmospheric correction [83]; which can be explained by enhanced scattering at red wavelengths negatively biasing NDVI values. Similarly, water vapor and thin cloud formations also negatively bias NDVI if unaccounted for by atmospheric correction as these constituents enhance absorption at near infrared wavelengths [84]; in turn reducing near infrared reflectance and NDVI. Assessment of sensor performance under low visibility conditions could improve our understanding of data usability with these atmospheric conditions.

Importantly, our findings suggest that SR products calculated from multispectral satellite sensors and on-the-go proximal sensors are highly correlated with each other and can be used interchangeably when assessing cover crop biomass and fractional groundcover. The various sensors produced high accuracy predictions of cover crop traits under clear-sky conditions and performed similarly. However, an NDVI intercept correction specific to active on-the-go proximal sensors could enable characterization of biophysical traits and eliminate the need for clear sky conditions. 

## 5. Conclusions

The results of this study demonstrate that SPOT-5, Landsat-7, Worldview-2, and CROPSCAN sensors were highly correlated with each other for both visible and near-infrared bands. CROPSCAN had high goodness of fit with cover crop biomass and groundcover (adj. *R*^2^ = 0.80 and 0.96; 56 kg ha^−1^ and RMSE = 5.52%, respectively). SR data were consistently closer to a 1:1 relationship with CROPSCAN than TOAR and were successfully used to estimate cover crop biomass (adj. *R*^2^ = 0.95–0.99, RMSE = 114.4–351.4 kg ha^−1^) and groundcover (adj. *R*^2^ = 0.95–0.99, RMSE = 2.37%–4.76%). The ability to adapt biomass and groundcover estimates across different platforms enables us to understand cover crop performance over larger areas and better link cover crop performance to environmental outcomes [2]. This has important implications for current efforts to map cover crop implementation and performance using imagery from multispectral satellite platforms. Data from passive and active proximal sensors can be collected as needed by farmers in clear or cloudy conditions and combined with satellite imagery to provide a more complete temporal understanding of cover crop growth and performance. This integration can aid in overcoming lingering issues of clouds and cloud shadows limiting the availability of data in the winter and spring or failing to be masked and negatively impacting data quality. 

## Figures and Tables

**Figure 1 sensors-24-02339-f001:**
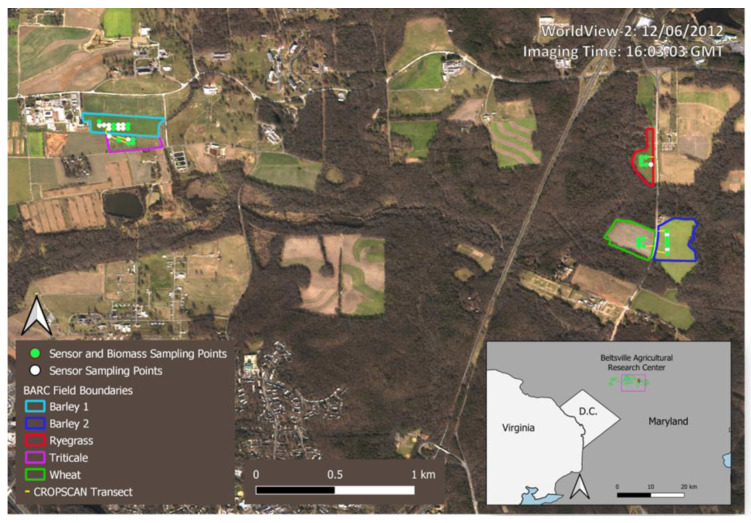
The study area consisted of five fields at the USDA-ARS Beltsville Agricultural Research Center (BARC) shown as the colored polygons on inset map. The proximal sensor transects overlay the white (multi-sensor) and green (multi-sensor and biomass) sampling points. Field locations and sampling points are shown on top of a WorldView-2 natural color image from 6 December 2012.

**Figure 2 sensors-24-02339-f002:**
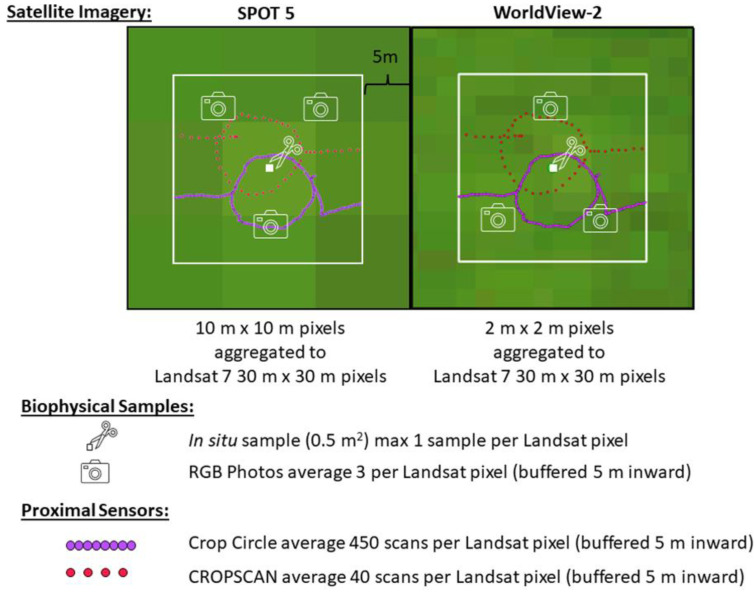
Diagram displaying the data types, collection, and processing for each sensor used in this study. For each of the three passive multispectral satellites (SPOT 5, Landsat 7, and WorldView-2), original and aggregated (Landsat 7) pixel sizes are represented, the biophysical sampling with both in situ samples and photos taken near the centroid of the Landsat 7 pixel, and the proximal data (active-Crop Circle, passive-CROPSCAN) collected inside of each Landsat 7 pixel buffered 5 m inwards.

**Figure 3 sensors-24-02339-f003:**
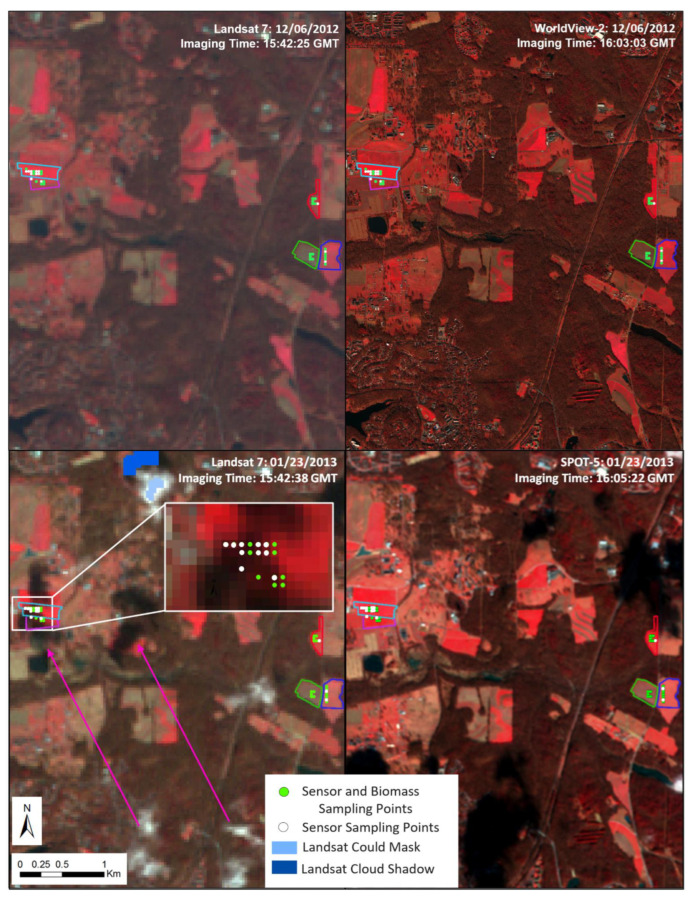
The study area is shown with four panels: 6 December 2012 Landsat imagery (collected at 15:42:25) in the upper left, 6 December 2012 WorldView-2 imagery (collected at 16:03:03, copyright 2012 Maxar) in the upper right, 23 January 2013 Landsat 7 imagery (collected at 15:42:38) in the lower left, and 23 January 2013 SPOT 5 imagery (collected at 16:05:22) in the lower right presented as a false color composite (near-infrared, red, green) with an overlay of white (multi-sensor) and green (multi-sensor and biomass) sampling points. The 23 January 2013, Landsat 7 inset shows sampling points obscured by cloud shadow. The pink arrows point out the location of clouds and their associated shadows. These areas were not detected using the cloud shadow and clouds mask that were included with the Landsat 7 Level-1 or the Landsat 7 Level-2 data products. The dark and light blue areas indicated in the legend are cloud and cloud shadows that were present in the Landsat 7 mask products.

**Figure 4 sensors-24-02339-f004:**
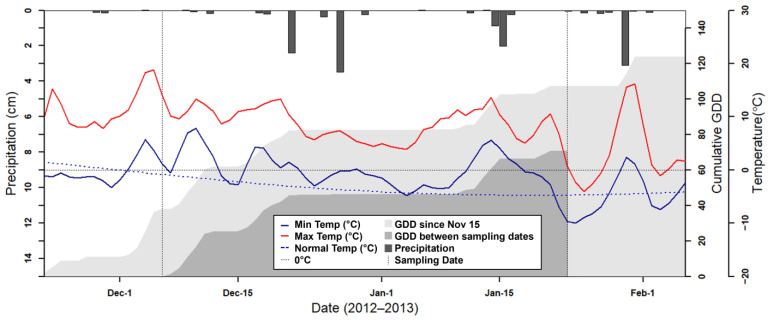
Climatic data for the study period show that there were approximately 70 accumulated growing degree days (dark gray) between the two satellite acquisition dates (vertical dotted lines), implying some minimal cover crop growth. The lighter gray represents accumulated growing degree days since 15 November, indicating the relative warmth of the cover crop growing season. The minimum temperatures (solid dark blue line) were slightly above climate normals (blue dotted line). Dotted horizontal line represents 0 °C. Data from a U.S. Department of Agriculture weather station at the Beltsville Agricultural Research Center, Beltsville, MD, USA.

**Figure 5 sensors-24-02339-f005:**
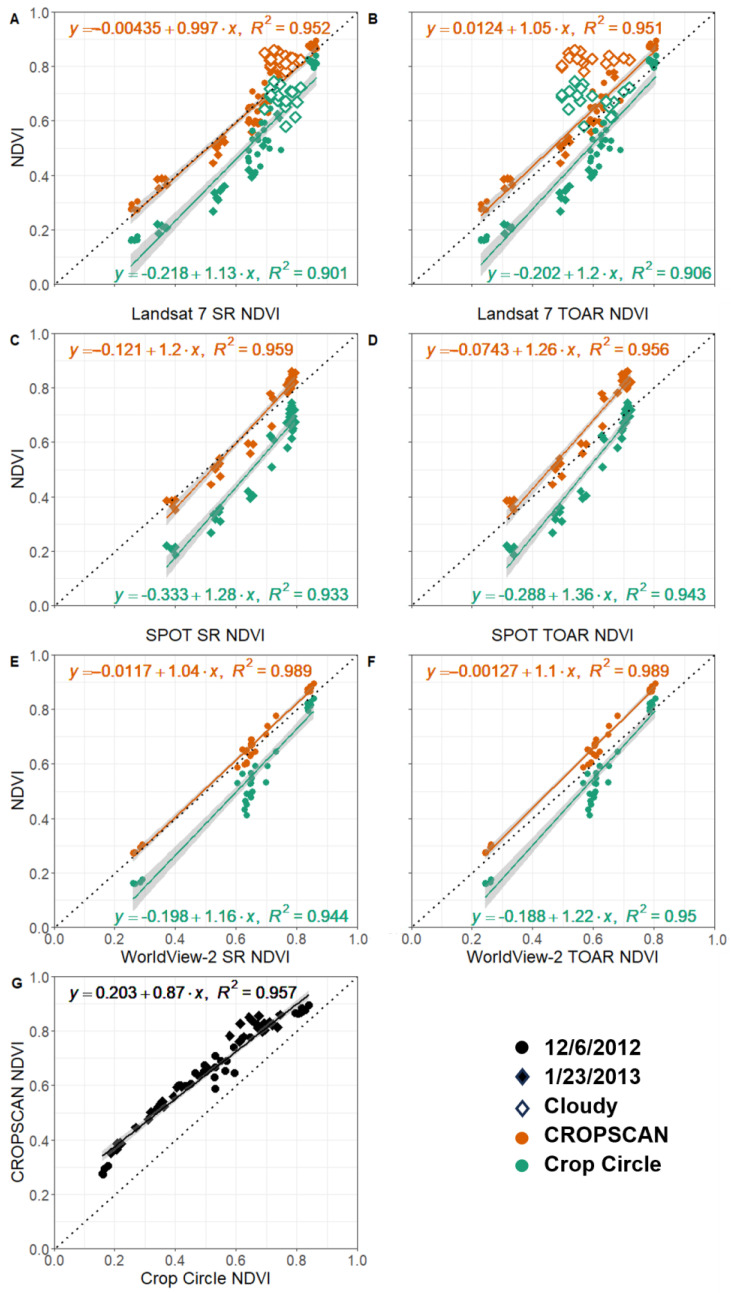
Linear regression of NDVI measurements from proximal sensors CROPSCAN (orange) and Crop Circle (teal) with satellite sensors Landsat 7 (**A**,**B**), SPOT 5 (**C**,**D**), and Worldview-2 (**E**,**F**) at two processing levels (surface reflectance [SR, (**A**,**C**,**E**)], top-of-atmosphere reflectance [TOAR, (**B**,**D**,**F**)]) on 6 December 2012 (circles) and 23 January 2013 (diamonds). The dashed line represents a 1:1 relationship with an intercept of zero. Panel (**G**) represents NDVI values from CROPSCAN and Crop Circle on both dates. The solid circles and diamonds are data points that were free of clouds. The hollow diamonds represent areas that are covered by cloud shadow in the 23 January 2013, Landsat 7 image and were excluded from the linear regression analysis. Linear regression and *R*^2^ values can also be found in Table 2.

**Figure 6 sensors-24-02339-f006:**
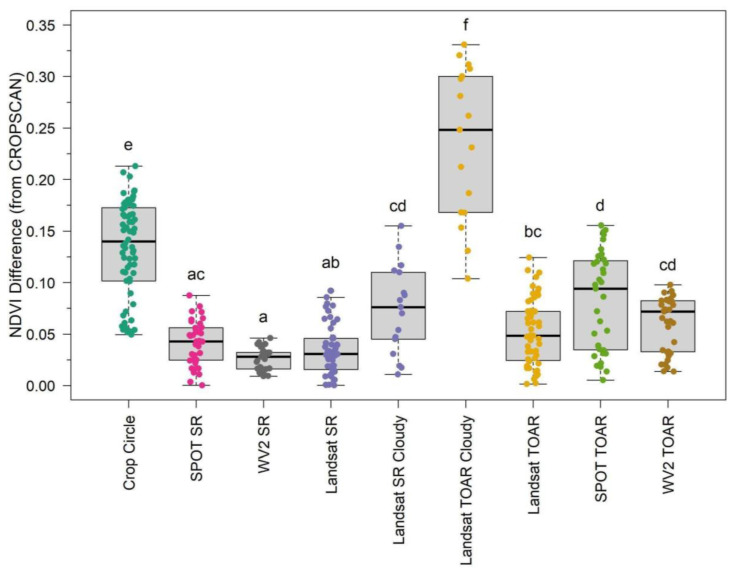
Boxplots showing the deviation in normalized difference vegetation index (NDVI) values from the passive proximal sensor. Statistical absolute value differences between groups (alpha < 0.05) are represented by letters above each sensor group. SR—surface reflectance, TOAR—top-of-atmosphere reflectance.

**Figure 7 sensors-24-02339-f007:**
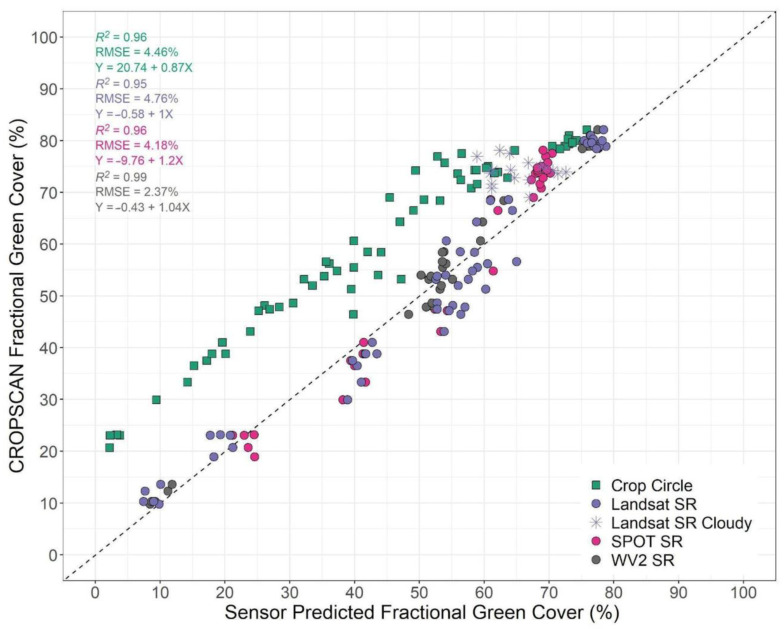
Linear regression of CROPSCAN fractional green cover and sensor-derived fractional green cover estimates. Sensors include Crop Circle (*n* = 70), Landsat 7 surface reflectance (SR) (*n* = 70), WorldView-2 (WV2) SR (6 December; *n* = 35) and SPOT 5 SR (23 January; *n* = 35). Sensor estimated fractional green cover was derived from the winter equation %GVC = −21.904 + 116.305 × NDVI described in Thieme et al. (2020) [18]. The dashed line represents a 1:1 relationship with an intercept of zero. Cloudy observations (*n* = 17) present in the Landsat 7 SR values (represented with a 
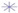
) are excluded from reported statistics on the left.

**Figure 8 sensors-24-02339-f008:**
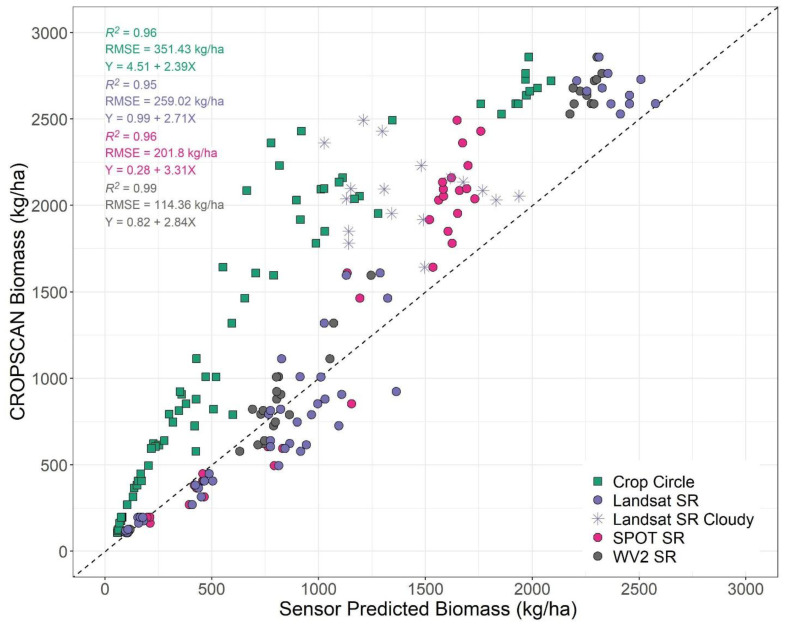
CROPSCAN-derived biomass compared to biomass derived from multiple sensors on 6 December 2012 and 23 January 2013. Sensors included are Crop Circle, Landsat 7 surface reflectance (SR), WorldView-2 SR (6 December) and SPOT 5 SR (23 January). Biomass was sampled on 14 December 2012 and 10 January 2013. Linear regressions were performed using estimated biomass derived from the following equation: ln(biomass) = 3.2022 + 5.3740 × NDVI for winter biomass described in Thieme et al. (2020) then delogged for the equations shown here [18]. The dashed line represents a 1:1 relationship with an intercept of zero.

**Table 1 sensors-24-02339-t001:** Specifications, collection times, red and near infrared (NIR) band ranges, and acquisition dates for proximal sensors, photos, and satellites. Times are listed in Coordinated Universal Time on T1 (first collection) and T2 (second collection).

Sensor	Landsat 7	WorldView-2	SPOT 5	CROPSCAN	Crop Circle	RGB Photos
Swath width (km)/footprint (m^2^)	185 km	16.4 km	60–80 km	1 m^2^	0.13 m^2^	2.7 m^2^
Repeat coverage	16 days	1.1–3.7 days	2–3 days	–	–	–
Altitude/Height	705 km	770 km	822 km	1.8 m	1 m	1.7 m
Ground resolution	30 m	1.84–2.4 m	10 m	–	–	–
Red band	630–690 nm	630–690 nm	610–680 nm	630–685 nm	660–680 nm	–
NIR band	770–900 nm	770–895 nm	780–890 nm	845–855 nm	775–810 nm	–
Date T1	6 December 2012	6 December 2012	–	6 December 2012	6 December 2012	14 December 2012
Start time T1	15:42:12.92	16:03:02.78	–	15:54:23	15:16:05.0	17:24
End time T1	15:42:39.68	16:03:04.39	–	17:53:41	17:27:07.8	18:19
Average data points per Landsat pixel T1	1	15 × 15	–	34	575	3
Date T2	23 January 2013	–	23 January 2013	23 January 2013	23 January 2013	23 January 2013
Start time T2	15:42:24.99	–	16:05:05.76	15:59:18	14:53:17.6	16:24
End time T2	15:42:51.75	–	16:05:43.6	17:23:08	18:11:21.6	18:40
Average data points per Landsat pixel T2	1	–	3 × 3	45	325	3

**Table 2 sensors-24-02339-t002:** Linear regression of red and near-infrared bands (NIR) and the normalized difference vegetation index (NDVI) values of sensors on 6 December 2012 and 23 January 2013. Sensors included are CROPSCAN, Crop Circle, Landsat 7 surface reflectance (SR), WorldView-2 SR, and SPOT 5 SR. For comparisons with data from both December and January, all samples were combined prior to analysis. Landsat 7 data from 23 January 2013 exclude pixels affected by clouds and cloud shadows.

Sensors Bands, *Date*	Slope	Intercept	adj. *R*^2^	RMSE
**CROPSCAN vs. Landsat 7 SR**				
Red, *6 December & 23 January*	1.199	−0.004	0.898	0.011
NIR, *6 December & 23 January*	1.141	−0.005	0.935	0.034
NDVI, *6 December & 23 January*	0.997	−0.004	0.952	0.041
**CROPSCAN vs. WorldView-2 SR**				
Red, *6 December*	1.080	−0.012	0.952	0.008
NIR, *6 December*	1.069	−0.019	0.951	0.030
NDVI, *6 December*	1.043	−0.012	0.988	0.020
**CROPSCAN vs. SPOT 5 SR**				
Red, *23 January*	1.674	−0.038	0.917	0.010
NIR, *23 January*	1.200	−0.004	0.925	0.031
NDVI, *23 January*	1.197	−0.121	0.959	0.036
**CROPSCAN vs. Crop Circle**				
Red, *6 December & 23 January*	0.659	0.018	0.804	0.015
NIR, *6 December & 23 January*	2.601	−0.386	0.770	0.061
NDVI, *6 December & 23 January*	0.870	0.203	0.956	0.038
**Crop Circle vs. Landsat 7 SR**				
Red, *6 December & 23 January*	1.771	−0.035	0.888	0.017
NIR, *6 December & 23 January*	0.344	0.180	0.866	0.015
NDVI, *6 December & 23 January*	1.130	−0.218	0.899	0.069
**Crop Circle vs. WorldView-2 SR**				
Red, *6 December*	1.522	−0.043	0.919	0.140
NIR, *6 December*	0.340	0.166	0.918	0.013
NDVI, *6 December*	1.159	−0.198	0.943	0.051
**Crop Circle vs. SPOT 5 SR**				
Red, *23 January*	1.880	−0.028	0.739	0.021
NIR, *23 January*	0.407	0.177	0.759	0.021
NDVI, *23 January*	1.281	−0.333	0.931	0.050
**Landsat 7 SR vs. WorldView-2 SR**				
Red, *6 December*	0.867	−0.002	0.959	0.006
NIR, *6 December*	0.917	−0.007	0.971	0.020
NDVI, *6 December*	1.019	0.004	0.977	0.028
**Landsat 7 SR vs. SPOT 5 SR**				
Red, *23 January*	1.099	−0.001	0.960	0.004
NIR, *23 January*	1.189	−0.032	0.976	0.009
NDVI, *23 January*	1.119	−0.071	0.984	0.016

**Table 3 sensors-24-02339-t003:** Linear regression of red and near-infrared bands (NIR) and the normalized difference vegetation index (NDVI) values of sensors on 6 December 2012 and 23 January 2013. Sensors included are CROPSCAN, Crop Circle, Landsat 7 top-of-atmosphere reflectance (TOAR), WorldView-2 TOAR, and SPOT 5 TOAR. For comparisons with data from both December and January, all samples were combined prior to analysis. Landsat 7 data from 23 January 2013 exclude pixels affected by clouds and cloud shadows.

Sensors Bands, *Date*	Slope	Intercept	adj. *R*^2^	RMSE
**CROPSCAN vs. Landsat 7 TOAR**				
Red, *6 December & 23 January*	1.415	−0.038	0.895	0.011
NIR, *6 December & 23 January*	1.203	−0.012	0.936	0.034
NDVI, *6 December & 23 January*	1.053	0.012	0.950	0.041
**CROPSCAN vs. WorldView-2 TOAR**				
Red, *6 December*	2.603	−0.042	0.951	0.008
NIR, *6 December*	2.323	−0.025	0.951	0.030
NDVI, *6 December*	1.099	−0.001	0.988	0.020
**CROPSCAN vs. SPOT 5 TOAR**				
Red, *23 January*	2.042	−0.096	0.916	0.010
NIR, *23 January*	1.298	−0.017	0.925	0.031
NDVI, *23 January*	1.261	−0.074	0.954	0.037
**Crop Circle vs. Landsat 7 TOAR**				
Red, *6 December & 23 January*	2.087	−0.085	0.884	0.017
NIR, *6 December & 23 January*	0.363	0.178	0.868	0.015
NDVI, *6 December & 23 January*	1.198	−0.202	0.905	0.067
**Crop Circle vs. WorldView-2 TOAR**				
Red, *6 December*	3.672	−0.086	0.920	0.014
NIR, *6 December*	0.741	0.164	0.920	0.012
NDVI, *6 December*	1.225	−0.188	0.948	0.049
**Crop Circle vs. SPOT 5 TOAR**				
Red, *23 January*	2.293	−0.093	0.739	0.021
NIR, *23 January*	0.440	0.172	0.759	0.021
NDVI, *23 January*	1.359	−0.288	0.941	0.046
**Landsat 7 TOAR vs. WorldView-2 TOAR**				
Red, *6 December*	1.793	0.000	0.960	0.005
NIR, *6 December*	1.887	−0.004	0.971	0.019
NDVI, *6 December*	1.016	−0.001	0.978	0.026
**Landsat 7 TOAR vs. SPOT 5 TOAR**				
Red, *23 January*	1.111	0.055	0.959	0.003
NIR, *23 January*	1.205	−0.034	0.976	0.009
NDVI, *23 January*	1.131	−0.049	0.987	0.013

## Data Availability

The raw data supporting the conclusions of this article will be made available by the authors on request.

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
