# Peer review of "Intercomparison of Same-Day Remote Sensing Data for Measuring Winter Cover Crop Biophysical Traits"

_sensors, 2024, doi:10.3390/s24072339_

Round 1

Reviewer 1 Report

Comments and Suggestions for Authors

Review 

In the study, the authors quantify the differences between three passive satellites (Landsat 7, WorldView-2, and SPOT 5) with two processing levels (surface reflectance and top-of-atmosphere reflectance). In addition, they quantify changes in sensor-derived estimates of cover crop biophysical traits (biomass and fractional groundcover) attributed to the various sensors. The paper is very useful and the findings may be useful in overcoming persistent problems associated with clouds and cloud shadows that limit data availability in winter/spring seasons or cannot be masked and negatively impact data quality.

1.     The paper title is too general.

Comment: I believe that the title does not fully reflect the essence of the work and requires adjustment.

2.     English language used in the paper is acceptable but requires some minor improvements.

Comment: Authors should avoid too long sentences, otherwise the report essence will be lost.

3.     The «Introduction» and «Materials and Methods» sections are presented clearly and in detail. Here is a detailed description of satellite sensor systems for displaying landscape surfaces and their comparative characteristics.

4.     The paper results are quite informative and does not raise doubts about its reliability. Here the authors give a detailed information about the reflectance comparisons of satellite and proximal sensors, comparisons of satellite processing level and proximal sensors, and sensor intercomparisons of cover crop biophysical traits.

5.     The conclusions quite fully reflect the goals set by the authors in the study.

In general, the paper contains interesting results in solving persistent problems associated with clouds and cloud shadows that limit data availability in winter and spring or cannot be masked and negatively impact data quality. I recommend accepting the paper for publication in the Sensors journal after correcting minor editing English in the text.

Comments on the Quality of English Language

English language used in the paper is acceptable but requires some minor improvements.

Reviewer 2 Report

Comments and Suggestions for Authors

This paper presents a thorough comparison of spaceborne and ground-based sensors performance for estimate crop vegetation parameters (i.e., NDVI, biomass). In principle, the experimental results are quite reliable given that different types of sensing approaches are evaluated (three satellites, active and passive proximal sensors) and that the experimental data are compared both with RGB photos and with in situ sampling. The motivations are well-presented and even the English quality is satisfactory.

In my opinion, the major concern is about the originality of the work, given that the comparison between proximal sensing and spaceborne acquisitions is not a novel topic in the scientific literature. I report some examples in the following:

·       Biney, James Kobina Mensah, et al. "Exploring the suitability of uas-based multispectral images for estimating soil organic carbon: Comparison with proximal soil sensing and spaceborne imagery." Remote Sensing 13.2 (2021): 308.

·       Wijesingha, Jayan, et al. "Comparison of spaceborne and uav-borne remote sensing spectral data for estimating monsoon crop vegetation parameters." Sensors 21.8 (2021): 2886.

·       Alexopoulos, Angelos, et al. "Complementary use of ground-based proximal sensing and airborne/spaceborne remote sensing techniques in precision agriculture: A systematic review." Agronomy 13.7 (2023): 1942.

·       Shuman, Craig S., and Richard F. Ambrose. "A comparison of remote sensing and ground‐based methods for monitoring wetland restoration success." Restoration Ecology 11.3 (2003): 325-333.

·       Wang, Sheng, et al. "Cross-scale sensing of field-level crop residue cover: Integrating field photos, airborne hyperspectral imaging, and satellite data." Remote Sensing of Environment 285 (2023): 113366).

Hence, I suggest the Authors to better stress the importance and the novelty of their contribution with respect to the existing literature.

Moreover, the Authors should add some considerations about the generalization of the obtained results, given that the presented activity refers to a fairly short testing period.

Other considerations are reported in the following:

·   ·     In Table 2 and 3 I suggest adding the considered day for the Cropscan vs Landsat and the Crop Circle vs Landsat comparisons.

·       ·     In Fig. 5 A-D the x-axis scale is missing.

·    ·     The utility of the destructive sampling carried out in situ does not emerge and it seems to add no useful information to the discussion. When the Authors say they take CropScan data as the 'gold standard' they should justify this choice more in depth, for example by showing the results of the comparisons between in situ sampling and all the other sensors. The same holds for the collected RGB photos.

Round 2

Reviewer 2 Report

Comments and Suggestions for Authors

The Authors adequately answered to the issues highlighted during the previous revision step. The manuscript quality is increased and it can be accepted in the present form.